# Seroprevalence of COVID-19 and Psychological Distress among Front Liners at the Universiti Malaysia Sabah Campus during the Third Wave of COVID-19

**DOI:** 10.3390/ijerph19116840

**Published:** 2022-06-02

**Authors:** Mohd Hanafiah Ahmad Hijazi, Mohammad Saffree Jeffree, Nicholas Tze Ping Pang, Syed Sharizman Syed Abdul Rahim, Azizan Omar, Fatimah Ahmedy, Mohd Hanafi Ahmad Hijazi, Mohd Rohaizat Hassan, Rozita Hod, Azmawati Mohammed Nawi, Sylvia Daim, Walton Wider

**Affiliations:** 1Faculty of Medicine and Health Sciences, Universiti Malaysia Sabah, Kota Kinabalu 88400, Sabah, Malaysia; hanafiah_hijazi@ums.edu.my (M.H.A.H.); syedsharizman@ums.edu.my (S.S.S.A.R.); azizan.omar@ums.edu.my (A.O.); fatimahmedy@ums.edu.my (F.A.); sylviadaim@ums.edu.my (S.D.); 2Faculty of Computing & Informatics, Universiti Malaysia Sabah, Kota Kinabalu 88400, Sabah, Malaysia; hanafi@ums.edu.my; 3Faculty of Medicine, Universiti Kebangsaan Malaysia, Cheras, Kuala Lumpur 56000, Selangor, Malaysia; rohaizat@ppukm.ukm.edu.my (M.R.H.); rozita.hod@ppukm.ukm.edu.my (R.H.); azmawati@ppukm.ukm.edu.my (A.M.N.); 4Faculty of Business and Communication, INTI International University, Nilai 71800, Negeri Sembilan, Malaysia; walton.wider@newinti.edu.my

**Keywords:** COVID-19, seroprevalence, psychological distress, front liner, university, psychological inflexibiliy, psychological mindedness, mindfulness, coping styles

## Abstract

In 2020, the COVID-19 pandemic struck the globe and disrupted various aspects of psychological wellbeing, more so in frontline workers. Research on assessing the seroprevalence of COVID-19 has been scarce; in addition, there are limited studies assessing the association between the seroprevalence of COVID-19 and psychological distress. Therefore, this study aimed to determine the seroprevalence of COVID-19 and the prevalence of psychological distress and to determine whether sociodemographic variables, occupational information variables, coping styles, and psychological processes might contribute to the development of psychological distress. A cross-sectional study involving 168 Universiti Malaysia Sabah (UMS) front liners was carried out to assess these issues. The Depression, Anxiety and Stress Scale (DASS-21) was employed to assess psychological distress, together with the COVID-19 Rapid Test Kit Antibody (RTK Ab) and a series of questionnaires, including a sociodemographic and occupational information questionnaire, the Balanced Index of Psychological Mindedness (BIPM) questionnaire, the Mindfulness Attention and Awareness Scale (MAAS), the Acceptance and Action Questionnaire (AAQ-II), and the Brief COPE questionnaire. The results demonstrated a seroprevalence of COVID-19 at 8.3% (95% CI = 5.0–14.0). Non-healthcare workers (HCWs) had a higher COVID-19 prevalence. Meanwhile, the prevalence of depression, anxiety, and stress among front liners was low (3.0%, 3.6%, and 1.2%, respectively). Younger people (aged 30 years old or less) and HCWs had a higher prevalence of psychological distress; being a HCW was significantly associated with a higher level of anxiety. Dysfunctional coping and psychological inflexibility were consistently found to be predictors for higher levels of the three psychological distress variables. This study suggested some alternatives that could be explored by mental health providers to address mental health issues among front liners at universities.

## 1. Introduction

The COVID-19 pandemic was declared worldwide in March 2020. The COVID-19 pandemic has caused more fatalities than the combination of the severe acute respiratory syndrome and Middle East respiratory syndrome outbreaks, even though the death rate for COVID-19 is relatively low [1]. Unfortunately, no effective cure has been developed [2]. As of 8 August 2021, the cumulative numbers of confirmed cases of COVID-19 was over 200 million, with over 4.2 million deaths globally since the start of the pandemic [3]. This risk was especially high amongst front liners. In addition to the conventionally recognized categories of healthcare workers (HCWs), there were other types of frontline workers who were also at increased risk occupationally of contracting COVID-19. This included include security services personnel, laboratory technicians, and student affairs staff [4]. American [5] and Singaporean studies have consistently demonstrated high levels of anxiety and depression amongst frontline workers [6]. It is hence crucial to protect the mental wellbeing of front liners [7], as previous epidemics such as SARS-CoV (2002) and MERS-CoV (2012) were shown to adversely influence mental health [8]. Early COVID-19 pandemic meta-analyses suggested the prevalence of stress at 29.6%, depression at 33.7%, and anxiety at 31.9% [9].

At the same time, there is little information about the true extent of COVID-19 seroprevalence in the community. Early studies in a similar population indicated healthcare workers had a higher incidence rate (IR) of being suspected COVID-19 cases (IR = 39.6/1000 population) compared to the general worker [10]. Nevertheless, serological detection of specific antibodies against SARS-CoV-2 could help to better estimate the true number of infections and inform public health interventions; recent studies demonstrated that the spread of COVID-19 exceeded what was reported based on real-time polymerase chain reaction (RT-PCR) testing [11]. Interestingly, COVID-19 does result in multiple neuropsychological sequelae; nevertheless, there are no studies looking at the association between serological outcomes and psychological implications.

Zhang et al. [12] asserted that it was crucial to determine factors associated with the psychological distress of front liners, with past studies reporting that psychological constructs, such as psychological flexibility, psychological mindedness, mindfulness, and coping styles, were among protective factors for psychological distress [13,14,15,16,17,18,19,20]. Psychological flexibility refers to flexible psychological reactions in order to confront distress and increase capacity to accept the present moment [21]. Amiruddin et al. [22] associated psychological flexibility with psychological mindedness, in which a person has the ability to be inquisitive about the thought processes of themselves and others. Mindfulness is another similar construct whereby a person is being attentive, aware, and openly accepting of the present moment [23]. Therefore, that psychological flexibility, psychological mindedness, and mindfulness potentially could improve individuals’ awareness with regard to internal psychological states, signals, and needs, which could protect against psychological distress. Furthermore, the literature reports that different coping styles in the face of traumatic experiences, such as natural disasters and pandemics, have an impact on individuals’ psychological distress; specifically, negative coping styles are associated with higher psychological distress [24,25]

Hence this study had the following specific objectives: (1) to determine the seroprevalence of COVID-19 among UMS front liners; (2) to determine the prevalence of psychological distress among UMS front liners during the COVID-19 pandemic; (3) to examine the association between the seroprevalence of COVID-19, sociodemographic variables, occupational information variables, coping styles, psychological processes, and psychological distress among UMS front liners during the COVID-19 pandemic.

## 2. Methods

### 2.1. Participants

The present study was performed at Universiti Malaysia Sabah (UMS) in Borneo, Malaysia from December 2020–June 2021. Accidental sampling techniques were deployed due to time constraints and a small total number of front liners. The name list of the front liners was obtained from the university staff registry, with all identified staff invited to participate in the present study. This included university hospital staff, the security division, the student affairs department, and medical laboratory technologists.

A serology test for COVID-19 was carried out using Standard Q COVID-19 IgM/IgG Combo by SD BIOSENSOR, Yeongtong-gu, Suwon-si, Gyeonggi-do (Seoul, Republic of Korea) for the seroprevalence study. This RTK Ab test kit tested for both types of antibody, i.e., immunoglobulin M (IgM) and immunoglobulin G (IgG) of the COVID-19 virus using a whole blood sample, and the result was readable after 15 min. This particular RTK Ab test kit produced a sensitivity of 91.7% and 79.2% for IgM and IgG, respectively (if tested after seven days of the initial PCR), and 100% for both immunoglobulins if tested after nine (IgM) and 12 days (IgG) of initial the PCR [26]. Positive COVID-19 RTK Ab test results were referred to the University Health Centre (UHC) for further management. Respondents who tested negative on the COVID-19 RTK Ab test, but showed symptoms as per the Annex 1 COVID-19 Management Guidelines in Malaysia No.5/2020, were also referred to the UHC for further management [27].

### 2.2. Measurement

Five questionnaires were used in this study on top of a standardized sociodemographic questionnaire: the Depression, Anxiety and Stress Scale (DASS), Brief COPE, the Balanced Index of Psychological Mindedness (BIPM), the Mindfulness Attention and Awareness Scale (MAAS), and the Acceptance and Action Questionnaire-II (AAQ-II). The sociodemographic and occupational factors consisted of the following items: age (as of 2021), type of job, sex, education level, and length of services. The Symptoms and Exposure History to COVID-19 questionnaire was based on Annex 1, *Garis Panduan Pengurusan COVID-19 di Malaysia No.5/2020, Kementerian Kesihatan Malaysia*. It included a subject’s history of being COVID-19 positive, clinical criteria, and epidemiological criteria.

The DASS is a set of self-reporting questionnaires consisting of three subscales measuring the emotional states of depression, anxiety, and stress [28]. It has internal consistencies (Cronbach’s alpha) of 0.91 for the depression subscale, 0.84 for the anxiety subscale, and 0.90 for the stress subscale [28]. The Malay DASS-21 equally enjoys reasonable internal consistency, with a Cronbach’s Alpha of 0.863 for the depression subscale, 0.850 for the anxiety subscale, and 0.837 for the stress subscale. The overall internal consistency is 0.940 [29].

The Brief COPE is a self-reporting instrument assessing how a person copes with a stressor. It contains 28 items examining the frequency of various coping methods [30], organized into problem-oriented, emotion-oriented, and dysfunctional coping subscales [31]. The Malay version of the Brief COPE [32] has good internal consistencies of 0.83, consistent with the original.

The BIPM is a self-reporting questionnaire that measures psychological mindedness (PM) and was originally developed by Nyclicek and Denollet [33]. The BIPM comprises 14 items in two subscales, namely interest (seven items) and insight (seven items) [34]. Interest is referred to as attending to one’s own internal feelings, while insight is referred to as understanding these feelings [34]. The seven-item interest and insight subscales of the BIPM shows good internal consistency with Cronbach’s alphas of 0.85 and 0.76, respectively. The Malay version of the BIPM [35] has good internal consistency with a Cronbach’s alpha score of 0.76 for the interest subscale, 0.75 for the insight subscale, and 0.79 overall.

The MAAS assesses individual differences in the frequency of mindful states, namely attention and awareness to the present moment [36]. MAAS is a self-reported single factor scale, consisting of 15 items on a Likert scale ranging from 1 (almost always) to 6 (almost never). The original MAAS has good consistency with a Cronbach’s alpha score ranging from 0.80 to 0.90 [37]. A Malay version of MAAS has strong internal consistency with a Cronbach’s alpha score of 0.851 [38].

The AAQ-II is an instrument developed to assess experiential avoidance and psychological inflexibility. Experiential avoidance refers to neglect or avoidance of unpleasant thoughts, bitter memories, or physical sensations leading to action against one’s values [39]. Psychological inflexibility is conceptualized as rigid or firm psychological reactions against one’s values to avoid distress, uncomfortable feelings, thoughts, and ignoring the present moment [40]. It is a unidimensional questionnaire on a seven-point Likert scale, ranging from 1 (never true) to 7 (very true). A higher AAQ-II score reflects higher psychological inflexibility, which is associated with higher levels of psychological problems. The original AAQ-II has a good internal consistency with a Cronbach’s alpha score of 0.88, and a good test-retest reliability over 3 and 12 months at 0.81 and 0.79, respectively [21]. The Malay version has good internal consistency with a Cronbach’s alpha score of 0.91 [40].

### 2.3. Data Analysis

The data were analyzed using the IBM Statistical Package for Social Sciences (SPSS) version 27 (developed by Norman H. Nie, Dale H. Bent, and C. Hadlai Hull, Chicago, IL, USA) [41]. All continuous data were described either using means and standard deviations (SDs) or medians (IQRs) depending on the normality of the data, whereas categorical data were described using frequencies and percentages (%) and were in binary form. Descriptive analysis (mean, median, mode, standard deviation, interquartile range) was employed to describe the age. The data collected for COVID-19 status, psychological distress, and other sociodemographic and occupational information variables, were displayed in categorical form. Analysis of the categorical data was performed by utilizing chi-square test or Fisher’s exact test. This test was performed to identify any significant differences between each psychological distress (depression, anxiety, stress) with COVID-19 RTK Ab status and sociodemographic and occupational information variables (except for age). Meanwhile, analysis of continuous data utilized the independent *t*-test or Mann–Whitney U test, examining for significant differences between the dependent variables and age, coping styles (problem-oriented, emotion-oriented, and dysfunctional), and psychological process (psychological mindedness, mindfulness, psychological inflexibility) variables.

## 3. Results

A total of 168 respondents out of 237 invited ones managed to complete both questionnaires and the COVID-19 RTK Ab blood test, giving a response rate of 70%. As shown in Table 1, the mean age of the respondents was 36 (SD = +7.00), with equal distribution by sex and type of job. The majority of respondents completed a tertiary level of education (64.3%) and had worked for four years or more (69%).

### 3.1. Seroprevalence of COVID-19 among UMS Front Liners

Based on Table 2, a total of 14 (8.3%, 95% CI = 5.0–14.0) respondents were found to have positive COVID-19 antibodies. Among those who had seroconversion, two (14.3%) were IgM positive, while nine (64.3%) were IgG positive. The rest were positive on both IgM and IgG (21.4%). Non-HCW frontliners had a higher prevalence of contracting the disease, as presented in Table 2. The prevalence of seroconversion, however, demonstrated a similar seroprevalence across age groups and sex (refer to Table 2).

### 3.2. Prevalence of Psychological Distress among UMS Front Liners

A chi-square or Fisher’s exact test was performed to test the relationship between sociodemographic variables, occupational information variables, and psychological distress. Based on Table 3, there was no significant relationship between age, sex, type of job, and psychological distress among UMS front liners.

### 3.3. Association between Psychological Distress and Seroprevalence of COVID-19

A chi-square or Fisher’s exact test was performed to test the association between psychological distress and seroprevalence of COVID-19. Based on Table 4, there was no significant relationship between psychological distress and the seroprevalence of COVID-19 among UMS front liners. Depression, anxiety, and stress had *p* values of 1.00, 0.45, and 1.00, respectively.

### 3.4. Association between Psychological Distress with Sociodemographic and Occupational Information Variables

Since the continuous variables were not normally distributed, a series of Mann–Whitney U tests were performed to look into the association between psychological distress and age, and Fisher’s exact test for sex, type of job, level of education, and length of working.

As presented in Table 5, there were no differences in term of psychological distress with regard to age. In addition, Table 6 and Table 7 indicated no significant differences for depression and stress, respectively, with regard to sex, type of job, level of education, and length of work. However, based on Table 8, it was found that HCWs had higher levels of anxiety compared to non-HCW. Meanwhile, there was no difference for anxiety with regard to sex, level of education, and length of working.

### 3.5. Association between Psychological Distress and Coping Styles

A series of Mann–Whitney U tests was performed to look into the association between psychological distress and coping styles, operationalized as problem-oriented, emotion-oriented, and dysfunctional coping styles. As presented in Table 9, the psychological distress components of depression, anxiety, and stress all had a significant association with the emotional-oriented and dysfunctional coping styles (*p* < 0.05). Specifically, respondents with abnormal levels of depression tended to use emotion-oriented and dysfunctional coping styles compared to respondents with normal levels of depression. In addition, problem-oriented coping styles were also significantly associated with anxiety (*p* < 0.05), but not depression and stress. Specifically, respondents with abnormal levels of anxiety tended to use problem-oriented coping styles compared to respondents with normal levels of anxiety.

### 3.6. Association between Psychological Distress and Psychological Process Variables

As per Table 10, a series of Mann–Whitney U tests was performed to evaluate the association between psychological distress and psychological process variables. The three psychological process variables evaluated were psychological mindedness, mindfulness, and psychological inflexibility. Depression, anxiety, and stress were all found to be significantly associated with psychological inflexibility. Specifically, respondents with abnormal level of psychological distress had higher psychological inflexibility compared to respondents with normal levels of psychological distress. However, neither psychological mindedness nor mindfulness had a significant association with the psychological distress variables.

## 4. Discussion

In examining the first research objective, the seroprevalence of COVID-19 among UMS front liners was 8.3%, which was higher compared to a previous study carried out in Malaysia that only involved HCWs [42]. This could be explained by the fact that this previous study was carried out at the early phase of Malaysia’s COVID-19 pandemic during the months of April and May 2020 and only involved HCWs who had no history of COVID-19 infection. It was similar to a recent meta-analysis (8.7% in HCWs) [43], but was much lower than the 35.4% reported in the first wave in the US [44]. This figure supported the concern of the underestimation of the COVID-19 burden, especially among front liners. When we stratified the respondents according to age, sex, and type of job, we found that being non-HCW had a higher prevalence of seroconversion. Meanwhile, age, and sex showed a similar seroprevalence of COVID-19, consistent with previous literature [44].

In examining the second objective, the prevalence of psychological distress was low compared to previous studies from Singapore (8.9%, 14.5%, and 6.6% for depression, anxiety and stress, respectively) [6] and China (44.4%, 46.0%, and 28.8%, respectively) [45]. The prevalence was also lower than the estimated prevalence of depression, anxiety, and stress in the general population at 29.6%, 33.7%, and 31.9%, respectively [9]. However, the first two studies were conducted on a larger number of respondents and hence might have reflected prevalence better. Furthermore, the study by [45], used a different case-finding instrument. This might also relate to the healthy worker effect [46] whereby exclusion of unhealthy individuals from employment markets artificially reduces the prevalence of ill health in workers. When we stratified the respondents according to age, sex, and type of job, there were no significant differences in psychological distress. Our findings were in contradiction with previous studies [47]; however, they were aligned with [19]. Therefore, our findings suggested that age, sex, or type of job were not protective factors for psychological distress.

There have been no previous studies replicating the insignificant relationship between psychological distress variables and seroprevalence of COVID-19. However, there were a few studies evaluating the association of history with COVID-19 infection and psychological distress, demonstrating higher scores for depression, anxiety, and stress [48] and a prevalence ranging between 47.0% to as high as 95.0% [49], which is in contradiction with the current findings. The reason there was no relationship between psychological distress variables with the seroprevalence of COVID-19 was probably imbalanced data. Furthermore, our findings suggested the type of job had a significant association on the prevalence of stress. HCWs experienced higher anxiety compared to non-HCWs, concurring with a previous study suggesting that many HCWs have higher anxiety, hence exposing themselves to increasing distress [50]. There were no significant differences in the prevalence of stress and depression according to marital status, sex, length of working, and educational level, in contradiction with previous literature [51,52].

Looking at the third research objective, emotion-oriented and dysfunctional coping styles demonstrated a significant association with all three psychological process variables. On the other hand, higher anxiety was demonstrated to be associated with problem-oriented coping styles, diverging from previous studies that showed positive effects of problem-oriented coping on mental health [53,54,55]. Dysfunctional and emotion-oriented coping styles contributed to psychological distress, dovetailing with predominant research [31,35,53,56,57,58]. Furthermore, only psychological inflexibility was found to be associated significantly with more psychological distress, which was consistent with the literature [59,60,61,62]. Conceptually, psychological flexibility can be understood as one’s higher-order or generalized ability to respond to situation demands effectively, in pursuit of longer-term goals, whereas psychological inflexibility is its opposite. The sudden and restrictive pandemic may have caused many individuals to not be able to draw on their usual responses or ways of coping and therefore default to behaviors that attenuate their stress in the short term, which led to more extensive avoidance in those lacking in flexibility. Meanwhile, those with greater psychological flexibility might have been relatively able to adapt to alternative and effective ways of responding.

## 5. Conclusions

This was the first study that explored the relationship between the seroprevalence of COVID-19 and psychological distress in a university population in Borneo, Malaysia. Although our study provided an important insight into the association between the seroprevalence of COVID-19 and psychological distress, the generalizability of the data might be limited due to the nature of accidental sampling, as mentioned above, and the small sample size. Thus, studies with more replicates are warranted. Despite low levels of psychological distress, importantly, this could be mediated by psychological flexibility; ultra-brief psychological intervention (UBPI) [35] incorporating acceptance and commitment therapy principles have been successfully employed to increase psychological inflexibility and general psychological wellness. There are many pathways for intervention that could have been adopted during the COVID-19 crisis. Creative solutions to the traditional face-to-face approach, complicated by social distancing rules and lockdowns, might include telecounseling and telepsychiatry, which have experienced a resurgence during the pandemic.

## Figures and Tables

**Table 1 ijerph-19-06840-t001:** Respondent profiles.

Variables	Frequency, *n* (%)
Age (years)	
Sex	
Male	81 (48.20)
Female	83 (51.80)
Type of job	
HCW	82 (48.80)
Non-health care worker	86 (51.20)
Education level	
Secondary education	60 (35.70)
Tertiary education	108 (64.30)
Length of services	
Less than four years	52 (31.00)
Equal to or more than four years	116 (69.00)

**Table 2 ijerph-19-06840-t002:** By age, by sex, and by type-of-job seroprevalence for COVID-19 status among UMS front liners.

Variables	COVID-19 Status					
	Negative, *n* (%)	Positive, *n* (%)	Overall, *n*	Chi Square	*p*-Value	Effect Size Phi/ Cramer’s V School
Age						
30 years old or less	34 (91.9)	3 (8.1)	37	0.03	0.96	0.04
Above 30 years old	120 (91.6)	11 (8.4)	131			
Sex						
Male	71 (87.7)	10 (12.3)	81	3.30	0.07	0.14
Female	83 (95.4)	4 (4.6)	87			
Type of job						
Healthcare worker (HCW)	79 (96.3)	3 (3.7)	82	4.58	0.03	0.17
Non-healthcare worker (non-HCW)	75 (87.2)	11 (12.8)	86			

**Table 3 ijerph-19-06840-t003:** By age, by sex, and by type-of-job prevalence of psychological distress.

Variables	Depression		Anxiety		Stress		Tot., *n*
	Normal, *n* (%)	Abnormal, *n* (%)	Normal, *n* (%)	Abnormal, *n* (%)	Normal, *n* (%)	Abnormal, *n* (%)	
Age							
30 years old or less	35 (94.6)	2 (5.4)	34 (91.9)	3 (8.1)	36 (97.3)	1 (2.7)	37
Above 30 years old	128 (97.7)	3 (2.3)	128 (97.7)	3 (2.3)	130 (99.2)	1 (0.8)	131
Sex							
Male	77 (95.1)	4 (4.9)	78 (96.3)	3 (3.7)	80 (98.8)	1 (1.2)	81
Female	86 (98.9)	1 (1.1)	84 (96.6)	3 (3.4)	86 (98.9)	1 (1.1)	87
Type of job							
Healthcare worker (HCW)	78 (95.1)	4 (4.9)	76 (92.7)	6 (7.3)	80 (97.6)	2 (2.4)	82
Non-healthcare worker (non-HCW)	85 (98.8)	1 (1.2)	86 (100.0)	0 (0.0)	86 (100.0)	0 (0.0)	86

**Table 4 ijerph-19-06840-t004:** Relationship between COVID-19 status and psychological distress variables.

COVID-19 Status (Antibody)	Anxiety (*n* = 168)				
	Normal, *n* (%)	Abnormal, *n* (%)	Total, *n*	X^2^	Sig. (Exact Sig.) ^b^
Positive	14 (100.0)	0 (0.0)	14	0.57	(0.45)
Negative	148 (96.1)	6 (3.9)	154		
	Depression (*n* = 168)				
Positive	14 (100.0)	0 (0.0)	14	0.47 ^a^	(1.00)
Negative	149 (96.8)	5 (3.2)	154		
	Stress (*n* = 168)				
Positive	14 (100.0)	0 (0.0)	14	0.18 ^a^	(1.00)
Negative	152 (98.7)	2 (1.3)	154		

^a^ two cells (50.0%) had an expected count of less than 5. The minimum expected count was 0.42; ^b^ *p* < 0.05.

**Table 5 ijerph-19-06840-t005:** Association between psychological distress and age.

Variables	Age (*n* = 168)			
	Mean Rank	Sum of Ranks	Z	Asymp. Sig. (Two-Tailed)
Depression				
Normal	85.06	13,865.00	−0.86	0.39
Abnormal	66.20	331.00		
Anxiety				
Normal	85.62	13,870.50	−1.55	0.12
Abnormal	54.25	325.50		
Stress				
Normal	84.94	14,100.50	0.28	0.32
Abnormal	47.75	95.50		

**Table 6 ijerph-19-06840-t006:** Association between depression and sociodemographic and occupational information variables.

Variables	Depression (*n* = 168)				
	Normal, *n* (%)	Abnormal, *n* (%)	X^2^	df	Sig. (Exact Sig.) ^b^
Sex					
Male	77 (95.1)	4 (4.9)	2.09 ^a^	1	(0.20)
Female	86 (98.9)	1 (1.1)			
Type of job					
HCW	78 (95.1)	4 (4.9)	2.01 ^a^	1	(0.20)
Non-HCW	85 (98.8)	1 (1.2)			
Level of education					
Secondary education	59 (98.3)	1 (1.7)	0.55 ^a^	1	(0.66)
Tertiary education	104 (96.3)	4 (3.7)			
Length of working					
Less than four years	50 (96.2)	2 (3.8)	0.20 ^a^	1	(0.65)
Equal to or more than four years	113 (97.4)	3 (2.6)			

^a^ two cells (50.0%) had an expected count of less than 5; ^b^
*p* < 0.05.

**Table 7 ijerph-19-06840-t007:** Association between stress and sociodemographic and occupational information variables.

Variables	Stress (*n* = 168)				
	Normal, *n* (%)	Abnormal, *n* (%)	X^2^	df	Sig. (Exact Sig.) ^b^
Sex					
Male	80 (98.8)	1 (1.2)	0.01 ^a^	1	(1.00)
Female	86 (98.9)	1 (1.1)			
Type of job					
HCW	80 (97.6)	2 (2.4)	2.12 ^a^	1	(0.24)
Non-HCW	86 (100.0)	0 (0.0)			
Level of education					
Secondary education	60 (100.0)	0 (0.0)	1.12 ^a^	1	(0.54)
Tertiary education	106 (98.1)	2 (1.9)			
Length of working					
Less than four years	51 (98.1)	1 (1.9)	0.34 ^a^	1	(0.53)
Equal to or more than four years	115 (99.1)	1 (0.9)			

^a^ two cells (50.0%) had an expected count of less than 5; ^b^
*p* < 0.05.

**Table 8 ijerph-19-06840-t008:** Association between anxiety and sociodemographic and occupational information variables.

Variables	Anxiety (*n* = 168)				
	Normal, *n* (%)	Abnormal, *n* (%)	X^2^	df	Sig. (Exact Sig.) ^b^
Sex					
Male	78 (96.3)	3 (3.7)	0.01 ^a^	1	(1.00)
Female	84 (96.6)	3 (3.4)			
Type of job					
HCW	76 (92.7)	6 (7.3)	6.26 ^a^	1	(0.01) ^b^
Non-HCW	86 (100.0)	0 (0.0)			
Level of education					
Secondary education	60 (100.0)	0 (0.0)	3.46 ^a^	1	(0.09)
Tertiary education	102 (94.4)	6 (5.6)			
Length of working					
Less than four years	50 (96.2)	2 (3.8)	0.02 ^a^	1	(1.00)
Equal to or more than four years	112 (96.6)	4 (3.4)			

^a^ two 2 cells (50.0%) had an expected count less than 5; ^b^
*p* < 0.05.

**Table 9 ijerph-19-06840-t009:** Association between psychological distress and coping styles.

Variables	Coping Styles (*n* = 168)											
	Problem-Oriented				Emotion-Oriented				Dysfunctional			
	Mean Rank	Sum of Ranks	Z	Asymp. Sig. (Two-Tailed)	Mean Rank	Sum of Ranks	Z	Asymp. Sig. (Two-Tailed)	Mean Rank	Sum of Ranks	Z	Asymp. Sig. (Two-Tailed)
Depression												
Normal	83.37	13,589.50	−1.72	0.09	83.23	13,567.00	−1.93	0.05 *	82.83	13,501.50	−2.54	0.01 *
Abnormal	121.30	606.50			125.80	629.00			138.90	694.50		
Anxiety												
Normal	83.05	13,620.50	−2.478	0.01 *	82.81	13,581.50	−2.88	0.00 *	83.08	13,625.00	−2.43	0.02 *
Abnormal	143.88	575.50			153.63	614.50			142.75	571.00		
Stress												
Normal	83.39	13,426.50	−1.417	0.16	82.87	13,342.00	−2.09	0.04 *	82.07	13,213.00	−3.11	0.02 *
Abnormal	109.93	769.50			122.00	854.00			140.43	983.00		

Note. * *p* < 0.05.

**Table 10 ijerph-19-06840-t010:** Association between psychological distress and psychological process variables.

Variables	Psychological Process Variables (*n* = 168)											
	Psychological Mindedness				Mindfulness				Psychological Inflexibility			
	Mean Rank	Sum of Ranks	Z	Asymp. Sig. (Two-Tailed)	Mean Rank	Sum of Ranks	Z	Asymp. Sig. (Two-Tailed)	Mean Rank	Sum of Ranks	Z	Asymp. Sig. (Two-Tailed)
Depression												
Normal	85.24	13,894.50	−1.13	0.26	84.47	13,768.00	−0.05	0.96	82.25	13,407.50	−3.44	0.00 *
Abnormal	60.30	301.50			85.60	428.00			157.50	788.50		
Anxiety												
Normal	84.30	13,826.00	−0.33	0.74	84.24	13,816.00	−0.44	0.66	82.66	13,555.50	−3.17	0.00 *
Abnormal	92.50	370.00			95.00	380.00			160.13	640.50		
Stress												
Normal	84.38	13,585.50	−0.15	0.88	84.49	13,603.00	−0.01	0.99	81.98	13,198.50	−3.25	0.01 *
Abnormal	87.21	610.50			84.71	593.00			142.50	997.50		

Note. * *p* < 0.05.

## Data Availability

Data can be made available upon reasonable request.

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
