# Peer review of "Seroprevalence of COVID-19 and Psychological Distress among Front Liners at the Universiti Malaysia Sabah Campus during the Third Wave of COVID-19"

_ijerph, 2022, doi:10.3390/ijerph19116840_

Round 1

Reviewer 1 Report

Dear authors.

Your work is very important and interesting. You made a good job, but it must be improved. I will try to show my comments and suggestions:

  1. Methods
    1. Participants
      1. You wrote "convenience sample". I do not agree. A convenience sample is when we want those participants in concrete. In this case, I suggest an accidental sample;
    2. Measurement
      1. You should explain why you used the variables "marital status and household income" which I do not find important for this study nor for the goals.
  2. Results
    1. You use too often quantifying terms, such as "most" which cannot be used without some statistical tests like binomial. Please, report the result without this judgement, only reporting the data. 
    2. When you describe the results of tables 2 and 3, the p-value, the effect size and power-test, are not available to support the description.  
    3. Why did you use the Shapiro-Wilk to analyse the normality when you had more than 50 participants in each distribution? Why did not you use the Kolmogorov-Smirnov with Lillefors correction which is done by the SPSS?
    4. page 7, line 237: you wrote: " higher among those who did not have seroconversion". This assumption is not supported by the data (table 5-p=1.00; table 6-p=0.45; table 7-p=1.0))
    5.  Data in 3.4.2.: please verify the data because everything what you say in this paragraph is not supported by the p-values in table 8;
    6. page 8, line 265: you wrote "based on table 11", but I think that you wanted to refer to table 10.
    7. Any comments/data to table 11 are available;
    8. Data in 3.4.3 | 3.4.4 
      1. In all analyses, the direction of the data (which category has the high score) and its mean rank are not available;
    9. Data in 3.5.1 | 3.5.2 
      1. You do not write anything about the regression model's characteristics (classification efficiency): the difference of overall percentage between the null model and the final regression model, the sensitivity and specificity and the AUC of the ROC curve
  3. Discussion | conclusions
    1. after the above improvements they must be changed
  4. Conflict of interest
    1.  You repeated twice the same declaration. (line 463)

Author Response

Dear Examiner, we are grateful for your consideration of this manuscript, and we also very much appreciate your suggestions, which have been very helpful in improving the manuscript. All the comments we received on this manuscript have been taken into account in improving the quality.

Reviewer 2 Report

i enjoyed reading your manuscript even though it’s a bit long. 
I wonder whether or not you care to mention any issues related to sample size, and if this may had affected your findings in any ways compared to other studies as it relates to stress, and depression.
Another questions is whether or not you are away of a similar study looking at similar variables in a population outside a university/hospital setting, and/or whether or not your population is representative of the background population in the area. 

Author Response

(The authors gave the same response as above.)

Reviewer 3 Report

The review of manuscript titled: Seroprevalence of COVID-19 and Psychological Distress among Front Liners in University Malaysia Sabah Campus during the Third Wave of Pandemic COVID-19 (ijerph-1691545)

Dear Authors,

this is interesting manuscript, a lot of work has been done, but in current form, the article cannot be accepted. I know it can be improved, but there are too much work to do.

Please, look into the following comments.

Abstract:

In my opinion some sentences does not sounds scientific, please paraphrase:

Here the first part of the sentence:

“COVID-19 pandemic has struck the globe like a storm in 2020 and causing various psychological distress to the world population especially the frontline workers, which will inevitably compromise the quality of services provided including health care and security”.

The assessment of psychological distress in relation to seroprevalence is a unexpected hypothesis. I would like to hear justification of the study.

The acronym “non-HCW” should be expanded.

Please, reduce the abstract, it is too long.

In the abstract there is a part of discussion, I would recommend to avoid such presentation like: Therefore, a specific health prevention promoting programme etc.

Please, underline accurately the aim of the study.

INTRODUCTION

The English language must be improved, there are some typo mistakes, as well, sentence constructions. Some of the are difficult to read like: To date. ,more fatalities than a combination of previous severe acute…….etc.

I suggest to change term frontilner for other one, maybe health care workers? and the define the group accurately in the methods?

I still do not see the casual relationship between IgE IgM antybodies and mental diseases. The anxiety, or depression might be induced in overworked health care workers, but I don’t think so by COVID-19 itself, at least this study does not prove that.

There are too many aims of the study. I would recommend to reduce the number of it, and rebuild manuscript. Authors are not able to control all the effects because the group study is too small.

I would suggest to create two manuscripts from the results and presented aims.

When the study was performed and in which period?

From the text I don’t know if health care workers were vaccinated or not?

Do the all participants fulfilled all questionnaire, it is hard to believe that there were willing and had time to do that.

Table 1 looks very awkward maybe, remove the mean and median of age? and input that information just into the text?

Gender is it not the same things as sex. The authors mention term “sex,” please, correct that.

3.1 Firs sentence probably correspond with total group, however, it is not underlined, so the reader must guess the intention of authors.

Construction of table 2 asks for chi2/fishers test.

Tables 3.

Norm, abnorm is incorrect, please paraphrase or add legend.

Table 4. Normality test is not necessary here and overdose the size of the manuscript, remove 3.3 part in my opinion.

There are too many tables in the manuscript, it should be reduced. It is difficult to read.

I don’t understand the results of table 15, please, explain. The title of the table for sure is incorrect.

Do authors mean occurrence of depression, anxiety etc?

The authors must propyl explain psychological flexibility and explain the interpretation of it.

The references list must be corrected according to journal standards, please, add information when authors accessed the internet sources. Like for ref 7. there should be mentioned information in brackets like: (accessed 11.05.2022) etc.

Author Response

Dear Examiner, we are grateful for your consideration of this manuscript, and we also very much appreciate your suggestions, which have been very helpful in improving the manuscript. All the comments we received on this manuscript have been taken into account in improving the quality.

Please find replies to your comments in the attachment. 

Round 2

Reviewer 3 Report

Dear Authors,

last comment, please, switch from gender term to sex. Your study is not sociological but from epidemiology field.

Kind regards,

Author Response

All gender term has been changed to sex.